# Host prion protein expression levels impact prion tropism for the spleen

**Vincent Béringue**[1]*, **Philippe Tixador**[1], **Olivier Andréoletti**[2], **Fabienne Reine**[1],
**Johan Castille**[3], **Thanh-Lan Laï**[1], **Annick Le Dur**[1], **Aude Laisné**[1], **Laetitia Herzog**[1],
**Bruno Passet**[3], **Human Rezaei**[1], **Jean-Luc Vilotte**[3], **Hubert Laude**[1]

**1** Université Paris-Saclay, INRAE, UVSQ, VIM Jouy-en-Josas, France, **2** Ecole Nationale Vétérinaire
Toulouse, INRAE, IHAP, Toulouse, France, **3** Université Paris-Saclay, INRAE, AgroParisTech, GABI, Jouy-
en-Josas, France

* vincent.beringue@inrae.fr

**Data Availability Statement:** All relevant data are within the manuscript and its supporting information files. Data are fully available without restriction.

## Abstract

Prions are pathogens formed from abnormal conformers (PrP$^{Sc}$) of the host-encoded cellular prion protein (PrP$^C$). PrP$^{Sc}$ conformation to disease phenotype relationships extensively vary among prion strains. In particular, prions exhibit a strain-dependent tropism for lymphoid tissues. Prions can be composed of several substrain components. There is evidence that these substrains can propagate in distinct tissues (e.g. brain and spleen) of a single individual, providing an experimental paradigm to study the cause of prion tissue selectivity. Previously, we showed that PrP$^C$ expression levels feature in prion substrain selection in the brain. Transmission of sheep scrapie isolates (termed LAN) to multiple lines of transgenic mice expressing varying levels of ovine PrP$^C$ in their brains resulted in the phenotypic expression of the dominant sheep substrain in mice expressing near physiological PrP$^C$ levels, whereas a minor substrain replicated preferentially on high expresser mice. Considering that PrP$^C$ expression levels are markedly decreased in the spleen compared to the brain, we interrogate whether spleen PrP$^C$ dosage could drive prion selectivity. The outcome of the transmission of a large cohort of LAN isolates in the spleen from high expresser mice correlated with the replication rate dependency on PrP$^C$ amount. There was a prominent spleen colonization by the substrain preferentially replicating on low expresser mice and a relative incapacity of the substrain with higher-PrP$^C$ level need to propagate in the spleen. Early colonization of the spleen after intraperitoneal inoculation allowed neuropathological expression of the lymphoid substrain. In addition, a pair of substrain variants resulting from the adaptation of human prions to ovine high expresser mice, and exhibiting differing brain versus spleen tropism, showed different tropism on transmission to low expresser mice, with the lymphoid substrain colonizing the brain. Overall, these data suggest that PrP$^C$ expression levels are instrumental in prion lymphotropism.

## Author summary

The cause of prion phenotype variation among prion strains remains poorly understood. In particular, prions replicate in a strain-dependent manner in the spleen. This can result

**Funding:** VB, JLV, OA and HL received grants from the European Network of Excellence NeuroPrion, from the GIS Prions (Groupement d'intérêt scientifique Prions). VB and PT received grants from Region Ile-de-France. VB and HR received grants from the French Medical Research Fondation (FRM, Equipe FRM (DEQ20150331689)). The funders had no role in study design, data collection and analysis, decision to publish, or preparation of the manuscript.

**Competing interests:** The authors have declared that no competing interests exist.

in prion asymptomatic carriers. Based on our previous observations that dosage of the prion precursor (PrP) determined prion substrain selection in the brain, we examine whether PrP levels in the spleen could drive prion replication in this tissue, due to the low levels of the protein. We observe that the prion substrain with higher PrP need for replication does barely replicate in the spleen, while the component with low PrP need replicates efficiently. In addition, other human co-propagating prions with differing spleen and brain tropism showed different tropism on transmission to mice expressing low PrP levels, with the lymphoid substrain colonizing the brain. PrP$^C$ expression levels may thus be instrumental in prion tropism for the lymphoid tissue. From a diagnostic point of view, given the apparent complexity of prion diseases with respect to prion substrain composition, these data advocate to type extraneural tissues or fluids for a comprehensive identification of the circulating prions in susceptible mammals.

## Introduction

Mammalian prions are proteinaceous pathogens causing fatal neurodegenerative diseases termed transmissible spongiform encephalopathies (TSE) in humans and animals. TSE include scrapie in sheep and goats, bovine spongiform encephalopathy (BSE) in cattle, chronic wasting disease (CWD) in cervids and Creutzfeldt-Jakob disease (CJD) in humans [1]. Prions are formed from abnormal, β-sheet enriched conformers (PrP$^{Sc}$) of the host-encoded cellular prion protein (PrP$^C$). Prions replicate by templating the conversion and polymerization of PrP$^C$ by an autocatalytic process [2, 3]. Multiple strains of prions are recognized phenotypically within the same host species. Strains are conformational variants of PrP$^{Sc}$, at the level of the tertiary and/or quaternary structure [4–6]. In the infected host, prion strains exhibit specific incubation periods, stereotyped clinical signs and neuropathology, and specific tropism for regions of the central nervous system (CNS) and the lymphoid tissue (for review [7, 8]). Thus, certain prions can replicate early and at fairly high levels in tissues of the lympho-reticular system such as the Peyer's patches, spleen and lymph nodes [9–11]. Lymphotropic prions are then transported from these early reservoirs of infectivity to the brain either by the enteric nervous system (Peyer's patches) or by the peripheral nervous system and the spinal cord (spleen, lymph nodes) [9, 12, 13]. After peripheral infection, the presence of differentiated follicular dendritic cells (FDC) is required for efficient prion replication in the lymphoid tissue and subsequent neuroinvasion [9, 14–20]. Yet, prions can differ in their capacity to neuroinvade. Certain prion strains can persist in the lymphoid tissue without accessing the CNS [21–24]. Of concern are BSE/variant CJD prions which are suspected to accumulate in human lymphoid tissue without neuroinvasion. 1:2000 exposed individuals in the UK may be silent carriers in the lymphoid tissue [25], causing risks of secondary transmission [26].

Although one dominant PrP$^{Sc}$ conformation is usually detected by conventional immunodetection methods, there is clear evidence that natural or experimental prion sources can be composed of several substrains in variable proportions [27–30]. As a result, experimental prion transmission, be it in a homotypic context (i.e. host PrP$^C$ and prion PrP$^{Sc}$ share identical primary sequence) or not, can lead to the isolation of different substrains in the brain and spleen of a single transgenically modified mouse expressing PrP [21, 22, 28, 31]. The reasons for such substrain segregation between brain and spleen and more generally for the incapacity of certain strains to replicate in the spleen remain poorly understood. It has been proposed that tertiary or quaternary PrP$^{Sc}$ conformations impact prion capacity to replicate in the spleen [24, 31–33].

Intracerebral transmission of natural sheep scrapie isolates referred to as LAN isolates [9, 34] to multiple lines of transgenic mice expressing ovine PrP$^C$ (VRQ allele at codons 136, 154 and 171 of the PrP-encoding gene, where V, R, and Q stand for valine, arginine, and gluta-mine, respectively) at variable levels revealed that PrP$^C$ expression levels in the brain critically determine prion substrain selection [29]. The so-called LA21K dominant component in LAN sheep brain propagated faithfully in the brain of mice expressing near physiological PrP$^C$ lev-els, while the so-called LA19K prion subcomponent phenotypically emerged in the brain of high expresser mice.

PrP$^C$ levels are ~20-fold lowered in the spleen compared to the brain, in both wild-type mice and ovine high expresser transgenic mice [21]. We thus interrogate whether PrP$^C$ expres-sion levels could impact prion (substrain) peripheralization. We compare the outcome of LAN transmission in the brain and spleen tissue of high expresser mice. The LA21K substrain pre-dominantly replicating on low expresser mice preferably colonized the spleen and the LA19K substrain with higher-PrP$^C$ level need failed to propagate there. Further, we show that a pair of co-existing substrains resulting from CJD adaptation to ovine high expressers, and exhibiting differing brain versus spleen tropism showed a similar PrP$^C$-dependent selection in transgenic mice expressing variable PrP$^C$ levels, the lymphotropic substrain replicating dominantly in ovine low expressers. Our findings raise the possibility that PrP$^C$ expression levels are instru-mental in prion tropism for the lymphoid tissue.

## Results

### PrP$^{res}$ is 19K-type in the brain but 21K-type in the spleen of high expresser tg338 mice intracerebrally inoculated with LAN sheep scrapie isolates

Tg338 mice overexpress the VRQ allele of ovine PrP. The PrP$^C$ levels in the mouse brain are ~8-fold higher than in the sheep brain [29]. The spleen-to-brain PrP$^C$ ratio is ~1:20 in tg338, as in wild-type mice [21]. We transmitted by intracerebral (IC) route 26 sheep scrapie isolates from the LAN group ([29], PrP$^{res}$ electrophoretic signature with unglycosylated fragments migrating around 21 kDa (21K-PrP$^{res}$, Fig 1A)) to high expresser tg338 mice and analyzed at the disease terminal stage the PrP$^{res}$ signature in brain and spleen by western blot. The country of origin and the genotype of the LAN isolates are detailed in S1 Table. For comparison, we used the PG127 sheep scrapie isolate (Fig 1A), which accumulates in both brain and spleen on transmission to tg338 mice, and exhibits a 21K-PrP$^{res}$ type in both tissues [35]. Replication of LAN isolates in tg338 mice resulted in different electrophoretic PrP$^{res}$ patterns in brain and spleen. All but two tg338 brains analyzed showed prominent accumulation of 19K-PrP$^{res}$ (Table 1; representative immunoblot in Fig 1A), pathognomic of LA19K phenotypic expres-sion, as previously reported [29]. In striking contrast, all the spleens analyzed exhibited a 21K-PrP$^{res}$ signature (Table 1; Fig 1A).

### PrP$^{res}$ is 21K-type in the spleen of high expresser tg338 mice intracerebrally inoculated with 19K "CH1641-like" isolates

We next wondered whether distinct molecular signatures would be observed in brain and spleen of tg338 mice infected with sheep scrapie isolates closely resembling to the reference scrapie strain CH1641 [36] (S1 Table). These isolates share a common 19K-PrP$^{res}$ signature in the natural host brain (Fig 1A, S1 Fig, [37, 38]). Because of the signature resemblance with BSE in sheep, such isolates were also termed "BSE-compatible" (S1 Fig, [39]). However, their trans-mission to tg338 mice led to isolation of prions in the brain with strain features identical to LA19K prions [39]. Replication of CH1641-like isolates in tg338 mice resulted in a 19K-PrP$^{res}$

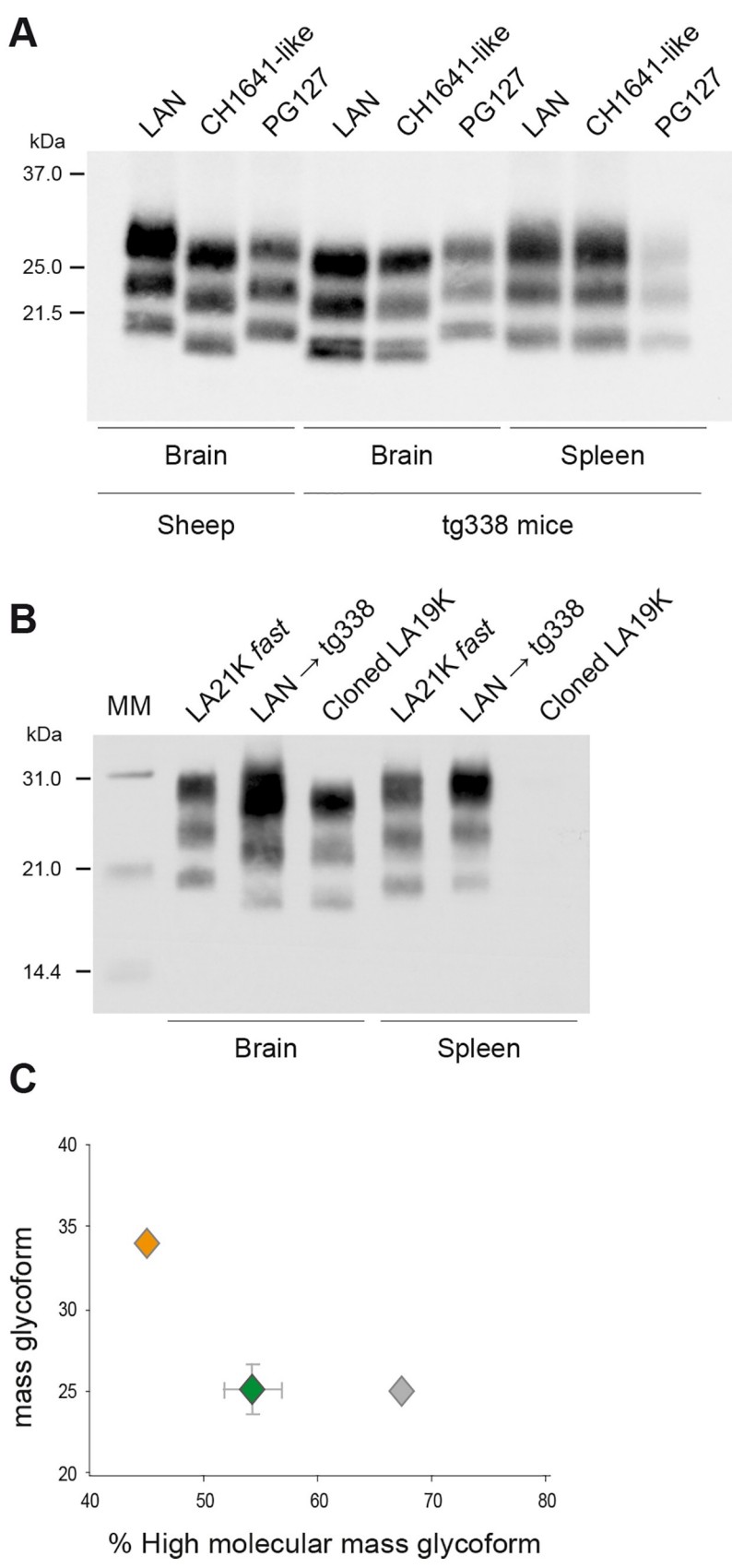

**Fig 1. PrP$^{res}$ electrophoretic pattern in brains and spleens of high expresser tg338 mice inoculated IC with LAN, CH1641-like isolates, and tg338-passaged LAN prions.** (A) PrP$^{res}$ profiles after challenge with representative LAN and CH1641-like isolates. The 21K-pattern observed after inoculation with PG127 sheep scrapie isolate is shown as control. The PrP$^{res}$ profile in the sheep brain is shown for comparison on the left side of the western blot. Note here and in the subsequent figures that 21K-spleen PrP$^{res}$ migrates slightly faster than 21K-brain PrP$^{res}$, as observed previously. Typically, spleen PrP$^{res}$ migration is 0.5–1 kDa faster than brain [10, 21, 28, 49, 51, 75]. (B) PrP$^{res}$ profiles after challenge with tg338-passaged LAN (6$^{th}$ passage; LAN → tg338), cloned LA19K and LA21K *fast* prions. MM: molecular mass markers. (C) Ratio of diglycosylated versus monoglycosylated PrP$^{res}$ in the spleen of mice after challenge with tg338-passaged LAN (gray symbol), LA21K *fast* (green symbol) and PG127 (orange symbol) prions (*n = 5* spleens analyzed at the 6$^{th}$ passage, data plotted as mean ± SEM).

signature in all the brains analyzed (Table 1; Fig 1A). In contrast, all but three spleens exhibited a 21K-PrP$^{res}$ signature (Table 1; Fig 1A). The remaining three spleens tested PrP$^{res}$-negative.

We thus observed a consistent, divergent 19K/21K PrP$^{res}$ signature in the brain/spleen on primary transmission of a large panel of LAN and CH1641-like isolates to tg338 mice, whatever the relative proportion of LA19K prions in the sheep brain inocula.

## The distinct PrP$^{res}$ types in brain and spleen are maintained on LAN serial passage

LAN serial passage (IC route) led to the dominant phenotypic expression of LA19K prions in tg338 brains ([29] and Fig 1B). In the spleen, the 21K-PrP$^{res}$ distinctive signature was conserved, up to the 6$^{th}$ passage (Fig 1B). This stably propagated 21K-PrP$^{res}$ signature in the spleen could arise from the replication of a strain type distinct from LA19K or from a tissue-specific proteolytic processing of LA19K prions [40]. To distinguish between these two possibilities, we analyzed the spleen colonization following IC challenge of tg338 mice with cloned LA19K prions which, at variance with LAN-passaged prions, do not co-exist with LA21K prions [29]. While 19K-PrP$^{res}$ was detected in the brain, the spleens of all terminally-sick animals tested negative (Fig 1B). Because of the limited sensitivity of the western blot, we inoculated reporter tg338 mice with these negative spleens to estimate the amount of infectivity. As shown in S2 Fig, the spleen extracts induced disease in 2 out of 8 mice >300 days post-infection, with a 19K-PrP$^{res}$ signature in the brain. By reporting these values to LA19K dose-response curve [6], we estimated that the spleens harbored 10$^5$-fold less infectivity than the brain. Thus, LA19K prions are mostly neurotropic and could not replicate in the spleen at levels detectable by western blot. Therefore, the 21K-PrP$^{res}$ signature observed in the spleen on serial passage of LAN originates from another strain component. It could be the LA21K component initially present in LAN brain or a highly pathogenic 'mutant' designated LA21K *fast* which occasionally

**Table 1. PrP$^{res}$ electrophoretic pattern in the brain and spleen after intracerebral inoculation of sheep scrapie isolates to tg338 mice.**

| TSE sources | Number [a] | PrP$^{res}$ type [b] | | |
|---|---|---|---|---|
| | | Sheep brain | tg338 brain (n/n$_0$) [c] | tg338 spleen (n/n$_0$) [c] |
| LAN | 26 | 21K | 19K (77/79) [d] | 21K (53/53) |
| CH1641-like | 11 | 19K | 19K (38/38) | 21K (18/21)[e] |
| PG127 | 1 | 21K | 21K | 21K |

[a]Number of isolates transmitted to tg338 mice

[b]As referred to the size of unglycosylated PrP$^{res}$ in immunoblots

[c]Number of mice with the PrP$^{res}$ type/number of mice tested

[d]Emergence of LA21K *fast* in two out of the 79 brains analyzed

[e]3 spleens tested PrP$^{res}$-negative

emerges on serial transmission of LAN to tg338 mice ([29], Table 1) and exhibits a 21K-PrP$^{res}$ signature in the spleen [41]. We thus compared the PrP$^{res}$ glycopatterns in tg338 spleens after inoculation with either LA21K *fast* or serially-passaged LAN prions. The LAN PrP$^{res}$ glycotype in the spleen was more abundantly glycosylated than that of LA21K *fast* (Fig 1B and 1C). It resembled that of LA21K prions in low expresser mouse brain [29] or sheep (Fig 1A). Prions accumulating in spleen on primary and serial passage of LAN are thus likely different from LA21K *fast* prions and resemble LA21K prions.

## LA21K prions replicate in the spleen of high expresser tg338 mice

To substantiate the view that the 21K-PrP$^{res}$ signature in the spleen is associated with LA21K replication, we compared the disease phenotype of reporter tg338 mice inoculated IC with spleen and brain containing tg338-passaged LAN prions. We also challenged intracerebrally low expresser tg143$^{+/-}$ mice (expressing 1.5-fold PrP$^C$ compared to sheep brain), as they allow the dominant propagation of LA21K prions in the brain [29]. The transmission data in the two mouse lines are summarized in Fig 2A. In tg338 mice, the mean incubation duration (ID) was >3-fold longer with the spleen than with the brain extract (413 ± 31 versus 129 ± 3 days). The spleen ID was also >2.5-fold longer than LA21K *fast* ID at the limiting dilution (157 days, [6, 29]), thus excluding the presence of this agent. The PrP$^{res}$ pattern found after inoculation of spleen material was 21K in all the brains and spleens analyzed (Fig 2A and 2B). The 21K pattern was confirmed by using the 21K-selective anti-PrP monoclonal antibody 12B2 ([42], Fig 2B, right panel). The strain-specified neuroanatomical deposition pattern of PrP$^{res}$ strikingly differed after inoculation of brain and spleen. In particular, numerous plaque-like PrP$^{res}$ deposits were seen specifically after inoculation of the spleen extract (Fig 2C). This pattern was reminiscent of LA21K prions [29]. Similarly, a LA21K phenotype was obtained on transmission of spleens from tg338 mice inoculated with other tg338-passaged LAN or CH1641-like isolates (S3 Fig). In tg143$^{+/-}$ mice, the spleen extract induced a long incubation time, as the brain (366 ± 12 days and 327 ± 52 days, respectively). A 21K-PrP$^{res}$ pattern was found in the diseased mice (Fig 2A and 2B). Such phenotype was reminiscent of LA21K prions in low expresser mice [29]. Together, these experiments suggest that LA21K prions replicate in tg338 mouse spleens on tg338-passage of both LAN and CH1641-like isolates.

## LA21K replicates as minor component in the brain of high expresser mice, allowing spleen colonization on LAN serial passage

Intracerebral inoculations, because of the spill-over of surplus inoculum outside the cranial cavity, usually result in systemic recirculation of prions and early colonization of the spleen [11, 35]. The constant detection of LA21K prions in tg338 spleens on serial passage of LAN, despite IC inoculation of brain material enriched in LA19K prions, is thus likely indicative of LA21K presence as subcomponent in the brain inoculum. This was also suggested by the LA21K phenotype obtained in low expresser tg143$^{+/-}$ mice inoculated with brain extract from tg338-passaged LAN ((Fig 2A and 2B) and [29]). To directly visualize LA21K prions in the brains of tg338 mice inoculated with LAN prions (5$^{th}$ passage), we performed a time-course analysis of PrP$^{res}$ accumulation in brain and spleen by using for the western blot analysis antibodies with differing reactivity toward 19K-PrP$^{res}$.

As shown in Fig 3 (bottom panel and quantification), 21K-PrP$^{res}$ accumulated in the spleen from day > 20 post-infection onwards and levels increased steadily until the disease terminal stage. In the brain, PrP$^{res}$ was detected from day 60 post-infection onwards and levels increased until the terminal stage of the disease (Fig 3, top panel). At day 60, the PrP$^{res}$ pattern was 21K in the brain. At day 100 and at terminal stage of the disease, 19K-PrP$^{res}$ was

**A**

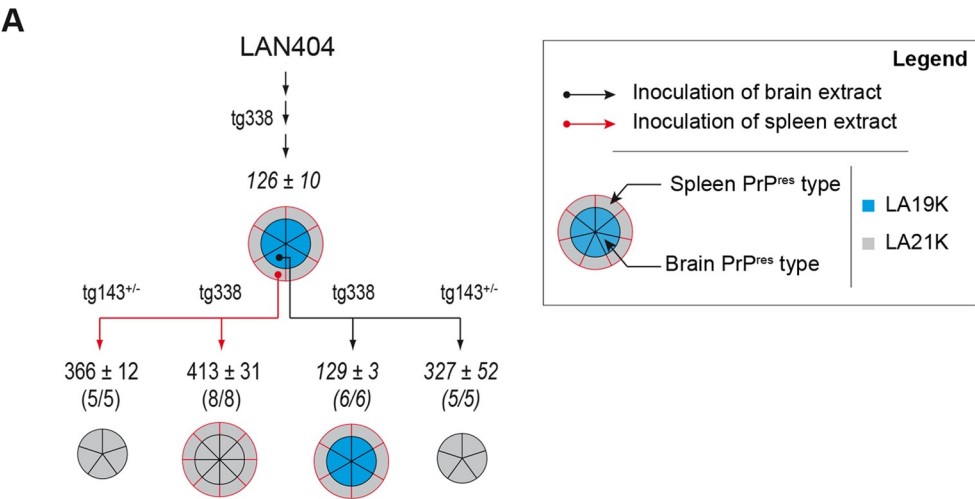

**B**

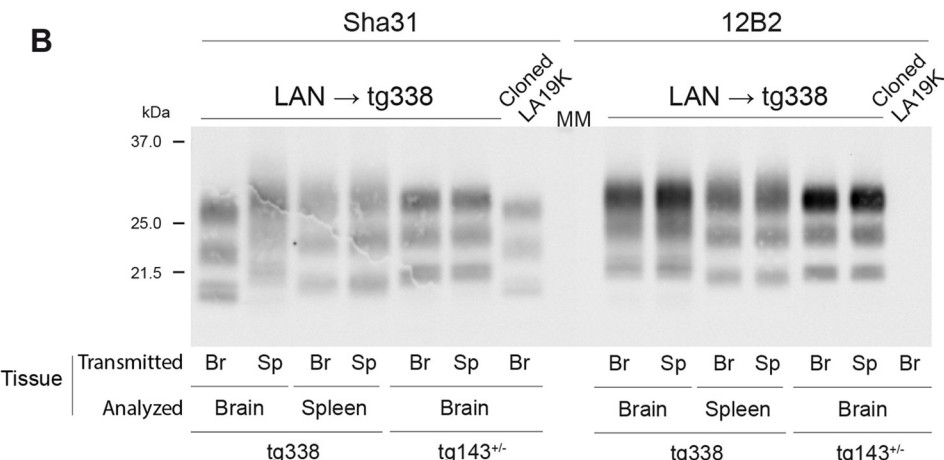

**C**

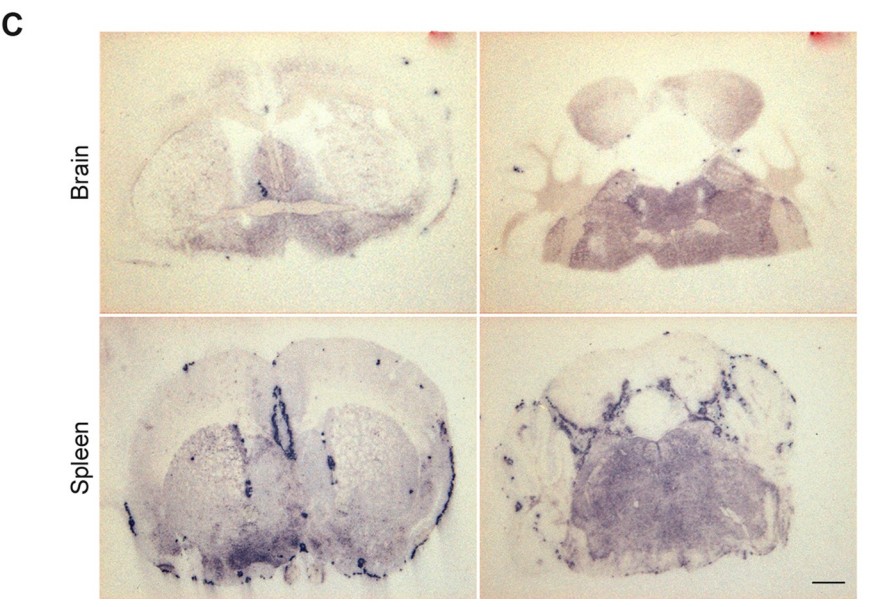

**Fig 2. Strain phenotype of prions replicating in tg338 mouse spleens on LAN serial passage.** (A) Summary of the transmissions (IC route) of brain or spleen extracts from tg338-passaged LAN (3rd passage, LAN → tg338) to high expresser (tg338) or low expresser (tg143+/-) mice. Transmission with brain or spleen extracts are indicated with black and red lines, respectively. The number of affected/inoculated mice and the mean survival times in days ± SEM are indicated for each inoculated group. Segmented, doubled circles are used to indicate the proportion of mice with dominant 19K PrP^res (blue) or 21K PrP^res signature (grey), either in the brain (inside of the circle, black lines) or in the spleen (outside of the circle, red lines). Data in italic are from [29]. Tg143+/- mice do not express PrP^C in the spleen, hence the absence of representation. (B) Representative immunoblot of PrP^res profiles in brain and spleen from high and low expresser mice, depending on whether tg338-passaged brain (Br) or spleen (Sp) material was used for inoculation. The immunoblots were revealed with Sha31 and 12B2 anti-PrP antibodies, as indicated. MM: molecular mass markers. (C) Representative histoblots at the level of the striatum (left panel) and the brain stem (right panel) showing the distribution of PrP^res deposits in the brains of high expresser tg338 mice inoculated with brain or spleen extracts. Scale bar, 1 mm.

dominant. The use of the 21K-selective anti-PrP monoclonal antibody 12B2 [42] to reveal the immunoblots demonstrated that 21K-PrP^res was still present in the brain at these time points (Fig 3, middle panel). Quantification of 12B2-positive PrP^res levels showed that 21K-PrP^res steadily accumulated over time in the brain. Collectively, these data suggest active co-replication of LA19K and LA21K in the brains of LAN-passaged tg338 mice, the 21K component being subdominant, except during the early phase of the replication.

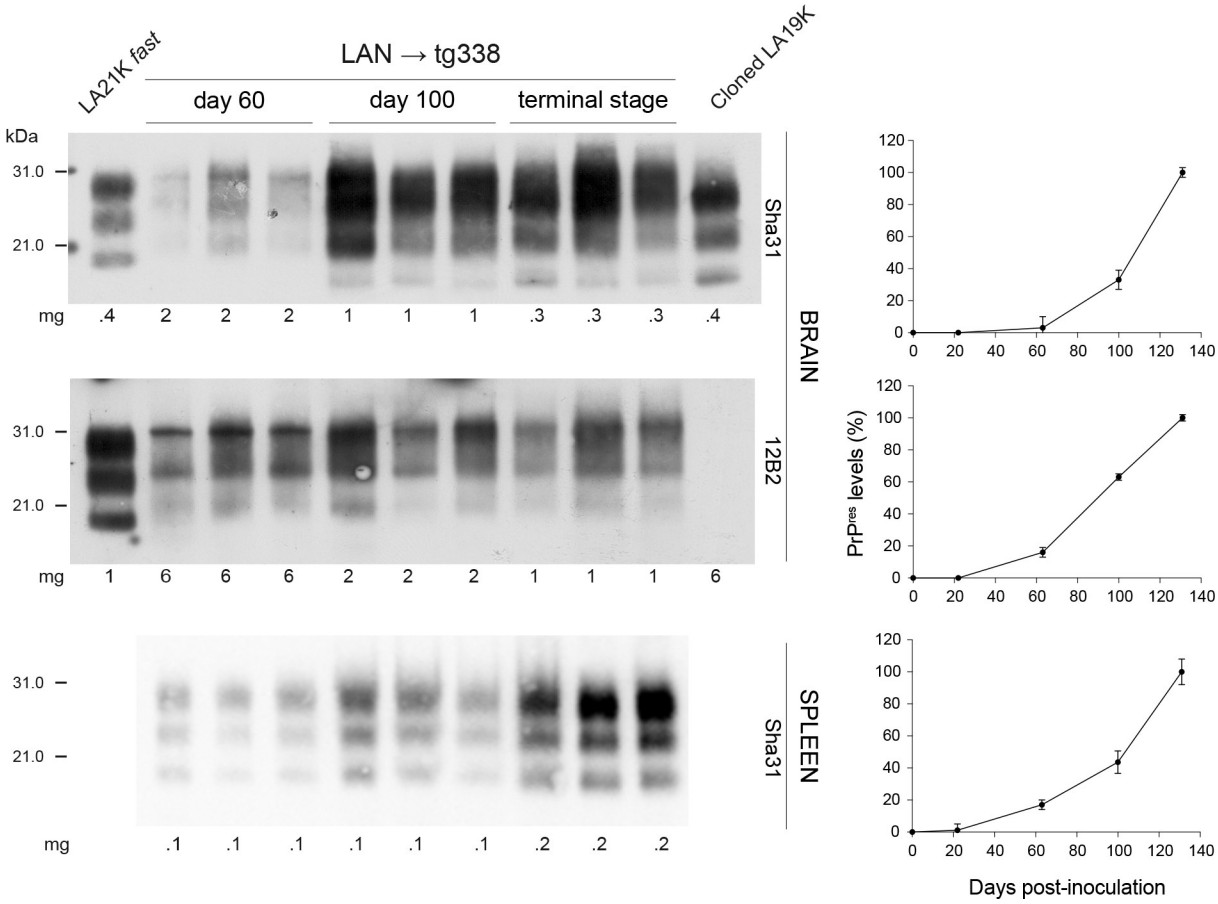

**Fig 3. Time course analysis of PrP^res accumulation in the brain and spleen of tg338 mice on LAN serial passage.** PrP^res profiles obtained after challenge with tg338-passaged LAN (5th passage; LAN → tg338). Three mice were analyzed at each time point. The immunoblots were revealed with Sha31 and 12B2 anti-PrP antibodies, as indicated. Cloned LA19K and LA21K *fast* PrP^res were used as 19K and 21K controls. The amount of mg tissue equivalent loaded in each lane is indicated. At each time point, PrP^res quantification was performed on mouse triplicates. Results are expressed as a percentage (± SEM) of the amount found in the ad hoc tissue at the terminal stage of disease.

To summarize this set of data, IC primary inoculation and serial passage of LAN and CH1641-like isolates to high expresser tg338 mice result in the preferential replication of the LA19K and LA21K subcomponents in the brain and spleen, respectively. In the brain, both LA19K and LA21K subcomponents coexist, with the LA19K subcomponent in higher proportion at the disease terminal stage. In the spleen, LA21K prions dominate, LA19K prions being barely able to replicate. LA19K prions thus show a preferential tropism for tg338 mouse brain. Such tissue-dependent segregation is consistent with the diverging selection of LA19K and LA21K prions on transmission to transgenic mice expressing PrP^C at varying levels [29].

## Enhanced propagation of LA21K prions in high expresser mouse brain on intraperitoneal inoculation of LAN and CH1641-like sheep isolates

Intraperitoneal (IP) inoculation favors the neuroinvasion of prions that primo-replicate in the spleen, by transport through the peripheral nervous system and the spinal cord (review [43]). We thus asked whether IP inoculation of LAN and CH1641-like isolates to tg338 mice would allow the dominant expression of LA21K prions in their brain as LA21K prions rapidly colonize their spleen. To provide a consistent picture, a group of representative sheep scrapie sources was transmitted to tg338 mice by IP route; four isolates were from the LAN group and 2 from the CH1641-like group. The PG127 isolate served for comparison. The dose injected was the same as for the IC route. Tg338 mice were euthanized at regular time-points post-injection and at the terminal stage of disease or end life to examine early spleen colonization and analyze by immunoblot the molecular PrP^res profile in both the spleen and the brain. In conventional mouse models inoculated IP with splenotropic prions, PrP^res is detected at early time-points in the spleen and the mean IDs are ~1.3-fold prolonged compared to IC infections at similar dose [44–47]. Similar features were found with the PG127 isolate regarding early PrP^res positivity in the spleen (from day 30) and ID prolongation (Fig 4A and 4B; S2 Table). With the LAN and CH1641-like sheep scrapie isolates, the spleens were uniformly and early 21K PrP^res-positive following IP inoculation (Fig 4A and 4B). Yet, the tempo and the clinical manifestation of the disease and the dominant strain type replicating in the brain were not uniform amongst the mice analyzed. The mean IDs in IP inoculated animals were on average ~3.5-fold longer than in the IC inoculated animals, ranging from 520 to 560 days (Fig 4A). The brain PrP^res patterns were heterogeneous with ~59% of 21K type and ~16% of 19K type (Fig 4A and 4B). The remaining ~24% tested PrP^res-negative. The 21K-PrP^res pattern dominated over the 19K-PrP^res pattern for all but one isolate (ARQ16). We conclude that IP inoculation of LAN and CH1641-like isolates favors the neuropathological expression of the splenotropic LA21K prions at the expense of LA19K prions. Nonetheless, the substantial proportion of PrP^res-negative mouse brains and the aberrantly prolonged IDs suggest a delayed pathogenesis.

## Detection of LA21K prions in high expresser mouse brain on intraperitoneal inoculation of tg338-passaged LAN prions

We next studied whether changing the IC to IP inoculation route similarly impacted the pathogenesis of tg338-passaged LAN prions. We inoculated the 4th and the 6th passage to analyze enough mice given the three different outcomes observed in the brain on primary passage (21K-PrP^res, 19K-PrP^res, PrP^res-negative). As for the primary isolates, all the spleens were 21K-PrP^res early, irrespective of the number of passage (Fig 4A and 4C), as after IC inoculation. In the brain, 21K-PrP^res was still detected. Yet, the proportion of 21K-PrP^res mice decreased at the expense of the 19K-signature with the number of passages. 71% and ~45% of the brains analyzed were 21K-positive at the 4th and 6th passage, respectively (Fig 4A and 4C).

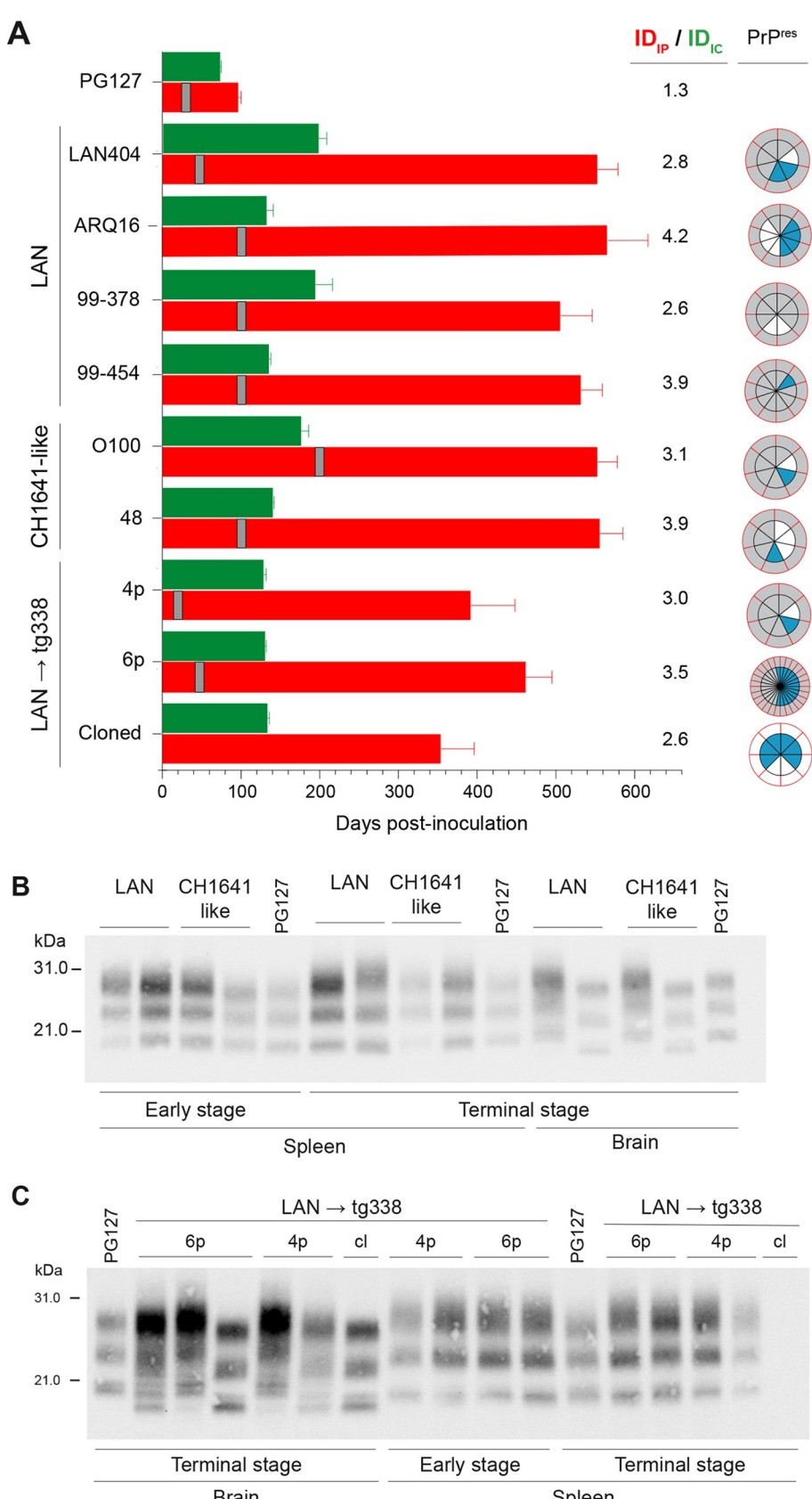

**Fig 4. PrP$^{res}$ electrophoretic pattern in brains and spleens of high expresser tg338 mice inoculated IP with LAN, CH1641-like isolates, and tg338-passaged LAN prions.** (A) Mean ± SEM incubation durations (ID) following challenge of tg338 mice by IC route (green bar) and IP route (red bar) with PG127, LAN, CH1641-like sheep scrapie isolates, tg338-passaged LAN (LAN → tg338, 4$^{th}$ and 6$^{th}$ passage (4p, 6p)) and cloned LA19K prions (Cloned). The ID$_{IP}$ / ID$_{IC}$ ratio are indicated. The grey vertical bar indicates the time at which PrP$^{res}$ was first detected in the spleen (S2 Table). Segmented, doubled circles are used to indicate the proportion of mice with 19K PrP$^{res}$ signature (blue), 21K PrP$^{res}$ signature (grey) or absence of PrP$^{res}$ (white), either in the brain (inside of the circle, black lines) or in the spleen (outside of the circle, red lines). (B) Representative immunoblot of PrP$^{res}$ profiles in brains and spleens from tg338 mice inoculated IP with isolates from the LAN and CH1641-like groups, at the terminal stage of disease. The spleen and the brain from the same animal are presented, meaning that the brain can harbor a 21K or 19K PrP$^{res}$ signature whereas a unique 21K signature is detected in the spleen. The spleen PrP$^{res}$ profiles are also shown at early time points post-inoculation (from 50 to 200 days depending on the isolate, S2 Table). Patterns after PG127 infection are shown for comparison. (C) Representative immunoblot of PrP$^{res}$ profiles in brain and spleen from tg338 mice inoculated IP with tg338-passaged LAN (LAN → tg338, 4$^{th}$ and 6$^{th}$ passage (4p, 6p)) and cloned LA19K prions (cl), at the terminal stage of disease. The spleen and the brain from the same animal are presented, meaning that the brain can harbor a 21K or 19K PrP$^{res}$ signature (with the 6$^{th}$ pass, a double 21/19K signature was observed), whereas a unique 21K signature is detected in the spleen. The spleen PrP$^{res}$ profiles are also shown at early time points post-inoculation (from 20 to 50 days depending on the number of passages, S2 Table). Patterns after PG127 infection are shown for comparison.

All but one remaining brains were 19K-positive. The mean IDs in IP inoculated animals were still 3.0 to 3.5-fold longer than in the IC inoculated animals (Fig 4A), suggesting no substantial evolution of the disease tempo despite the enrichment in LA19K prions over passaging (in both the inoculum used for IP inoculation and the diseased mice brains).

## Direct neuroinvasion of cloned LA19K prions after intraperitoneal inoculation

A substantial proportion of mice were 19K-PrP$^{res}$ positive in the brain after IP inoculation of LAN prions, despite the limited replication of LA19K prions in tg338 spleens. This suggested that LA19K prions could directly neuroinvade. To address this, we studied the pathogenesis of cloned LA19K prions after IP inoculation. Such pathway of infection resulted in a disease at near full attack-rate, with 19K-PrP$^{res}$ accumulating in the positive brains, whereas all the spleens remained PrP$^{res}$-negative (Fig 4A and 4C). These findings indicate that LA19K prions could directly neuroinvade from the peripheral sites of infection and be expressed phenotypically in the brain, a pathway previously observed for other prions in situation of impaired replication in the spleen [19, 20, 48]. To be noted, the IP ID was ~2.6-fold longer than the IC ID (Fig 4A), i.e. a value within the range observed with uncloned material.

## Inoculation of high expresser mice with spleen from LAN infected sheep results in predominance of LA21K prions in the brain

We asked whether a similar distinctive splenotropism between LA21K and LA19K prions pre-existed in sheep naturally infected with LAN. A spleen extract from the LAN404 isolate was inoculated by IC and IP routes to tg338 mice. At variance with the IC-inoculation of LAN404 brain material which resulted in dominant expression of LA19K prions in the brain and LA21K in the spleen in ~200 days (Table 1, Figs 1A and 5A), IC-inoculation of LAN404 spleen material led to accumulation of prions with an abundantly glycosylated 21K-PrP$^{res}$ signature in both brain and spleen in ~450 days (Fig 5A and 5B). The 21K pattern obtained was confirmed by using the 21K-selective anti-PrP monoclonal antibody 12B2 (Fig 5B, right panel). These phenotypic features were resembling those obtained after IC transmission of tg338 spleens infected with tg338-passaged LAN/CH1641-like prions (Figs 2 and S3) and suggested that LA21K prions were recovered from sheep spleen transmission.

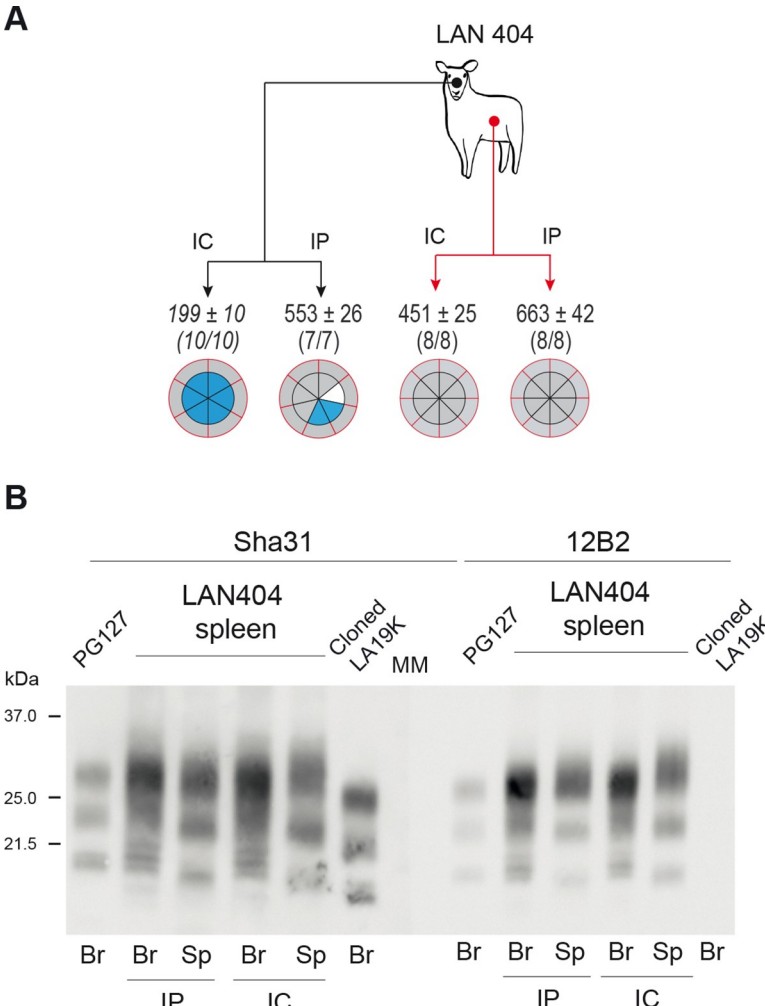

**Fig 5. Strain phenotype of prions replicating in tg338 mouse spleen and brain on primary passage of LAN404 spleen.** (A) Summary of the transmissions by IC or IP route of brain or spleen extracts from LAN404 sheep scrapie isolate to tg338 mice. Legends are as in Fig 2 and Fig 4. Data in italic are from [29]. (B) Representative immunoblot of PrP^res electrophoretic pattern in (Br) and spleen (Sp) from high expresser tg338 mice, upon inoculation with LAN404 spleen material by IP or IC route. PrP^res electrophoretic patterns of PG127 and cloned LA19K in tg338 mouse brain are shown for comparison. The immunoblots were revealed with Sha31 and 12B2 anti-PrP antibodies, as indicated. MM: molecular mass markers.

After IP inoculation, the incubation time established at 660 days. At variance with IP inoculation of LAN404 brain material which resulted in different outcomes (21K-PrP^res, 19K-PrP^res, PrP^res-negative), all the brains were positive and exhibited a 21K-PrP^res signature (Fig 5A and 5B). All the spleens were positive with a 21K-PrP^res signature. This suggested again dominant expression of LA21K prions. To be noted, the mean ID in IP inoculated animals was ~1.5-fold longer than in the IC inoculated animals. The IC to IP ID lengthening was thus less pronounced than after challenge with LAN/CH1641-like brains and resembled that observed in conventional mouse models.

We conclude that in the spleen sheep, the LA21K component predominates over the LA19K component to levels allowing 100% molecular expression in the CNS. This predominance of LA21K in the sheep spleen also allowed faster pathogenesis on IP inoculation as compared to brain.

## Another pair of strains co-existing in high expresser mice and with differing tropism for the brain and spleen shows divergent evolution in low expresser mice

To further explore the possibility that prion capacity to replicate in the spleen is controlled by PrP$^C$ expression levels, we compared the replicative capacity in low expresser mice of a pair of co-existing strains with opposite splenotropism in high expressers. These strains, termed T1$^{Ov}$ and T2$^{Ov}$, were isolated by adapting cortical MM2 sporadic CJD (MM2-sCJD) prions to tg338 mice. T2$^{Ov}$ replicated dominantly in tg338 brain with T1$^{Ov}$ being present as a subcomponent, while T1$^{Ov}$ preferentially populated tg338 spleen ([28] and Fig 6A and 6B). A brain extract from tg338-passaged MM2-sCJD prions was IC inoculated to tg335$^{+/-}$ mice, expressing 1.2-fold PrP$^C$ compared to sheep brain [29]. For comparison, brain extracts containing either cloned T2$^{Ov}$ prions or cloned T1$^{Ov}$ prions [28] were inoculated to tg335$^{+/-}$ mice. The transmission of tg338-passaged MM2-sCJD prions to tg335$^{+/-}$ mice led to the dominant phenotypic expression of T1$^{Ov}$ prions, as based on the mean ID (Fig 6A), the PrP$^{res}$ profile in brain and spleen by immunoblotting (Fig 6B) and the neuroanatomical distribution of PrP$^{res}$ by histoblotting (Fig 6D), which all showed the T1$^{Ov}$ characteristics observed after infection with cloned T1$^{Ov}$ prions (Fig 6A–6D). The apparent counter-selection of T2$^{Ov}$ prions in tg335$^{+/-}$ mice was not due to their incapacity to replicate as cloned T2$^{Ov}$ prions elicited disease in these mice in ~330 days (Fig 6A) with a T2$^{Ov}$-PrP$^{res}$ specific banding pattern in the brain (Fig 6B) and presence of low levels of PrP$^{res}$ by histoblotting (Fig 6D). Together, these data highlight, for another pair of co-propagating prion strains, a tight correlation between capacity to replicate in the spleen and capacity to be dominantly expressed in mice expressing low PrP$^C$-levels in the brain.

Remarkably, the dominant expression of T1$^{Ov}$ prions on passage of tg338-passaged MM2-sCJD prions to tg335$^{+/-}$ mice was not accompanied by a subdominant replication of T2$^{Ov}$ prions; over two back-passage to tg338 mice, T2$^{Ov}$ prions were not phenotypically rescued (Fig 6A). The T1$^{Ov}$ signature remained dominant in all brains and spleens from tg338 mice (Fig 6A and 6C). Alternance of prion transmission to mice expressing variable levels of PrP$^C$ may thus achieve a degree of selection akin that in a heterotypic PrP transmission context.

## Discussion

Building on our previous observations that the phenotypic dominance of two prion substrains co-existing in variable proportions in certain sheep scrapie isolates is driven by PrP$^C$ expression levels in transgenic mice, we now show that the segregation between these two substrains occurs between the brain and the spleen of the same infected high expresser mouse, these tissues expressing different PrP$^C$ levels. In addition, a pair of co-existing substrains derived from CJD and exhibiting differing brain versus spleen tropism in high expresser mice shows divergent expression in low expresser mice, with unique expression in the brain of these mice of the splenotropic substrain. Host PrP$^C$ levels may thus impact prion (in)capacity to replicate in the lymphoid tissue.

LA19K prions phenotypically predominated in high PrP$^C$ expresser tg338 mouse brains inoculated IC with a panel of natural sheep scrapie isolates termed LAN and composed in variable proportion of LA19K and LA21K prions ([29] and this study). In the spleen of these mice however, we show early, widespread, and dominant accumulation of LA21K prions, -as based on molecular strain typing, and phenotypic characterization on high- and low-PrP$^C$ expresser mice. The LA19K/LA21K segregation in brain/spleen was also observed after IC inoculation of CH1641-like isolates in which the LA19K subcomponent was originally dominant. The

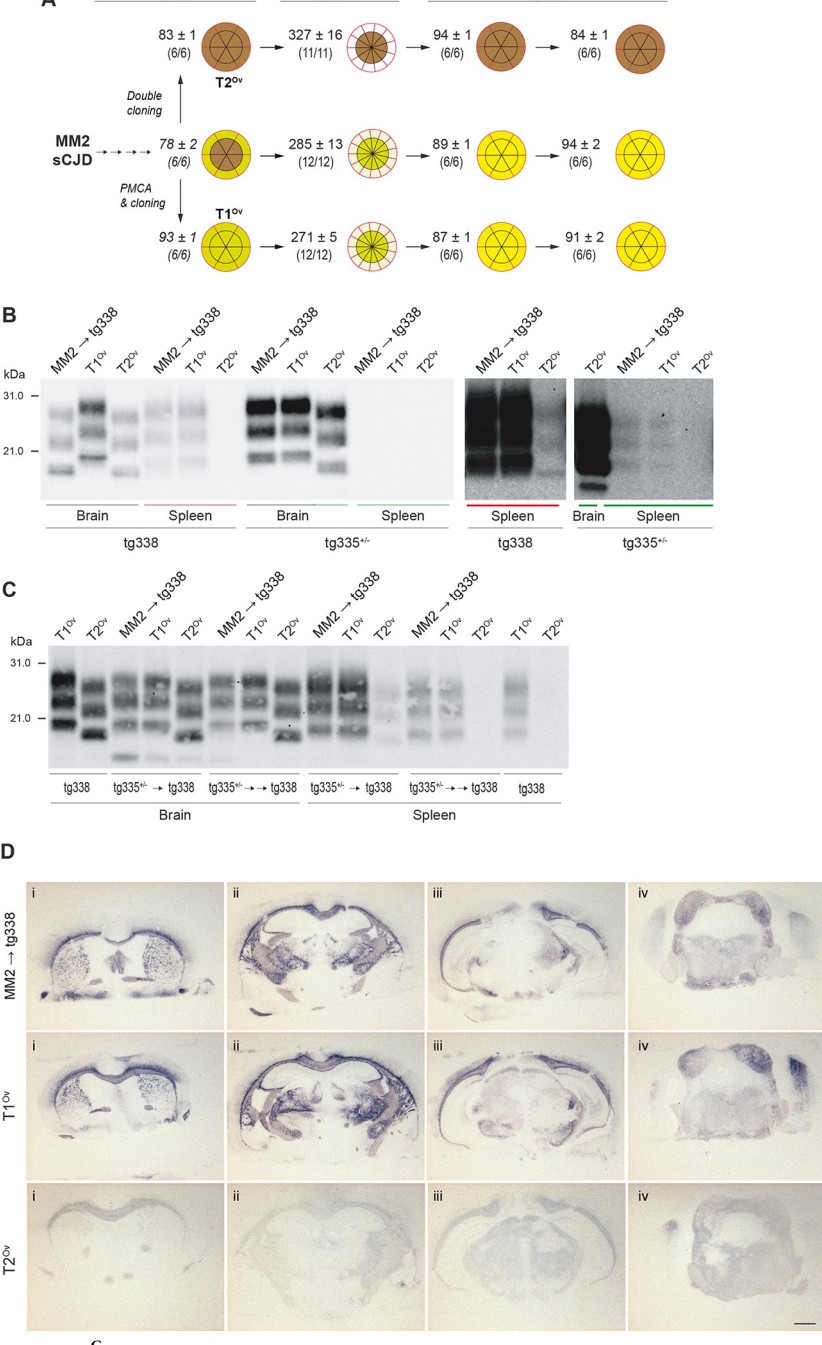

**Fig 6. PrP^C-level dependent selection of prions with differing splenotropism in ovine PrP transgenic mice.** (A) Summary of the transmissions (IC route) of brain extracts from tg338-passaged MM2-sCJD prions (4th passage), cloned T1^Ov or T2^Ov prions on low expresser tg335^+/- mice, before back passage to tg338 mice. The mean survival time in days ± SEM and the number of affected/inoculated mice (mice with TSE and positive for brain PrP^res by immunoblot) are indicated for each inoculated group. Segmented, doubled circles are used to indicate the proportion of mice with T2^Ov (brown or clear brown depending on the accumulation levels) or T1^Ov (yellow or clear yellow) PrP^res signatures in their brains (inside of the circle, black lines) and spleens (outside of the circle, red lines). Data in italic are from [28]. (B) Representative immunoblot of PrP^res profiles in brain and spleen from tg338 high expresser mice and tg335^+/- low expresser mice infected with tg338-passaged MM2-sCJD (MM2 → tg338), cloned T1^Ov or cloned T2^Ov prions. The gels on the right side are portions of the gels (colored red and green lines) that are purposely overexposed to detect T2^Ov PrP^res in the spleen. (C) Representative immunoblot of PrP^res profiles in brain and spleen from tg338 high expresser mice after one or two back-passage from tg335^+/- low expresser mice infected with tg338-passaged MM2-sCJD (MM2 → tg338), cloned T1^Ov or cloned T2^Ov prions. PrP^res signatures from T1^Ov and

T2$^{Ov}$ prions in tg338 mice are shown for comparison. (D) PrP$^{res}$ deposition pattern in the brain of tg335$^{+/-}$ low expresser mice infected with tg338-passaged MM2-sCJD, cloned T1$^{Ov}$ or cloned T2$^{Ov}$ prions. Representative histoblots of antero-posterior coronal brain sections at the level of the septum (i), hippocampus (ii), midbrain (iii) and brainstem (iv). Scale bar, 1 mm.

lymphoid tissue was thus a privileged site for LA21K prions replication. LA19K prions replicated with limited efficacy in the spleen, as shown by transmission of cloned LA19K. LA19K prions were thus considered as mostly neurotropic.

That co-existing TSE substrains replicate preferentially in distinct tissues from the same host is likely to constitute a general observation (i.e., not limited to laboratory animals nor to specific polymorphisms in the PrP gene). We show here a similar enrichment of LA21K prions in the spleen relative to brain in the natural sheep host. A distinct selection of PrP$^{res}$ conformers with 19K and 21K signature was also observed in the brain and spleen tissue from transgenic mice expressing the ARQ allele of ovine PrP on inoculation with CH1641-like scrapie isolates [49]. A similar tissue-specific segregation of prion substrains due to distinct tropism was observed on transmission of a human vCJD case to transgenic mice overexpressing human PrP [22], on transmission of cortical MM2-sporadic CJD to tg338 mice [28], directly in a sporadic CJD individual [50] and in CWD-infected elk [51].

A methodological implication of our findings is that strain typing in both brain and spleen tissues is required for a comprehensive identification of the TSE agents circulating in susceptible individuals. Blind strain typing of LAN isolates in tg338 mouse brain would categorize them as CH1641 prions. However, co-examining the tg338 lymphoid tissue or changing the route of infection allowed phenotypic expression of the LA21K strain component. *In fine* transgenic mice in which PrP is introduced by additive transgenesis prove to be a valuable tool to reveal natural prion strain diversity present in TSE isolates, provided the PrP-encoding constructs allow correct PrP$^C$ expression in the lymphoid tissue. From a diagnostic point of view, comprehensive strain typing is an important issue given the apparent complexity of natural animal and human TSE isolates in terms of substrain heterogeneity [27, 52, 53].

The determinants of prion replication in the lymphoid tissue remain mostly unknown. From a prion structural viewpoint, two broad hypotheses recently emerged, involving PrP$^{Sc}$ tertiary or quaternary structure. Shikiya et al. suggested that the sensitivity to degradation of the PrP$^{Sc}$ assemblies determines prion tropism for the spleen, as based on the absence of DY prions replication in hamster spleens and the sensitivity of DY PrP$^{Sc}$ to protease digestion [32]. Here, we firmly exclude this possibility; lymphoincompetent cloned LA19K prions and lymphocompetent tg338-passaged PG127 prions (127S strain, [35, 54]) were equally resistant to proteolysis, either in vitro (PK resistance) or ex vivo (clearance by peritoneal macrophages) (Fig 7). Aguilar-Calvo et al. proposed that prion aggregation size impacts prion neuroinvasion after peripheral infection, based on the observation that highly fibrillar prion assemblies are poor neuroinvaders but high replicators in the spleen, as compared to subfibrillar prions [31]. PrP$^{Sc}$ assemblies' particulate profile may indeed impact access to the FDC from the site of infection [55, 56]. LA21K prions would follow this rule as they can be categorized as 'highly fibrillar', as based on the presence of plaque-like deposits when replicating dominantly in the brain (Fig 2D). However, T1$^{Ov}$ and T2$^{Ov}$ prions, which exhibit similar 'subfibrillar' aggregation size [57] showed markedly differing capacity to replicate in the spleen (Fig 6). Thus, such a hypothesis cannot fully explain our observations.

From the host viewpoint, mature FDC are necessary for prion replication [9, 14–20, 58, 59], being ontogenetically present or induced by chronic inflammation [60]. Co-factors, including notably complement or complement receptors [61–63], have been proposed to be involved.

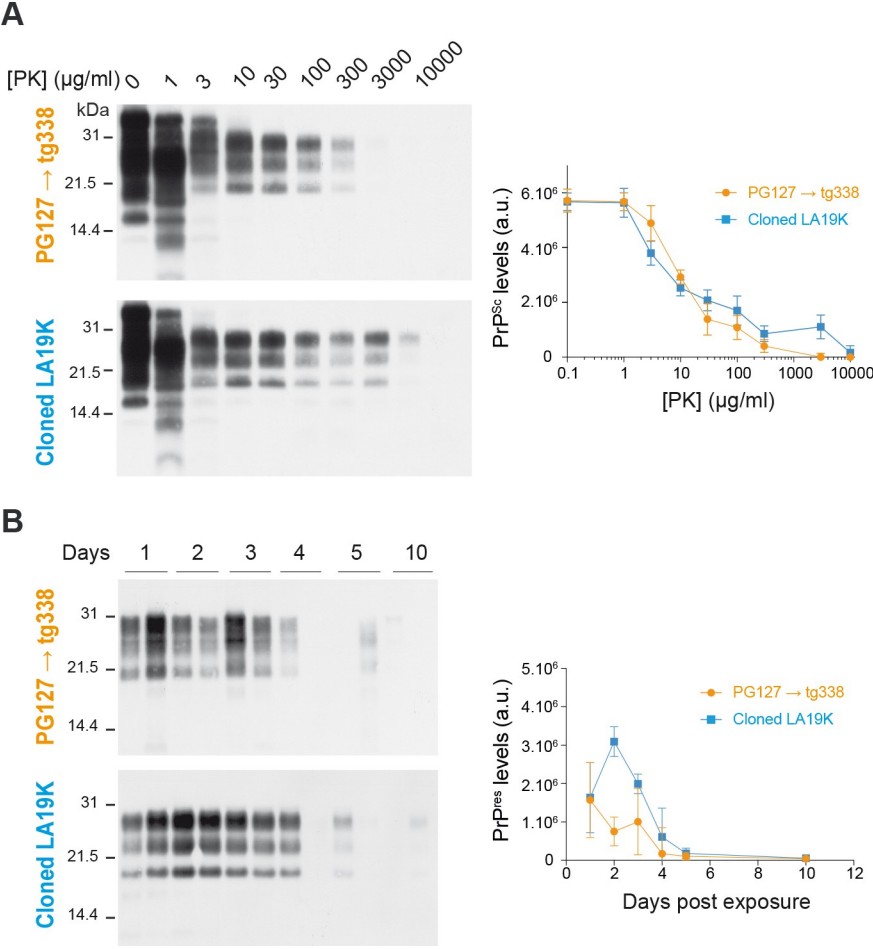

**Fig 7. Resistance of LA19K prions to proteolysis.** (A) Comparative PK digestion assay performed on brain material of tg338 mice inoculated with tg338-passaged PG127 prions (PG127 → tg338) or cloned LA19K prions. The homogenates were treated with increasing concentrations of PK, as indicated, and analyzed by western blotting. PrP$^{Sc}$ quantification was performed on $n$ = 5 independent experiments. Results are expressed as total PrP$^{Sc}$ levels (arbitrary units (a.u.) ± SEM). (B) Representative immunoblot showing PrP$^{res}$ degradation in tg338 peritoneal macrophages exposed to tg338-passaged PG127 (PG127 → tg338) prions or cloned LA19K prions for the indicated period (duplicates analysis). PrP$^{res}$ quantification is expressed as total PrP$^{res}$ levels (arbitrary units (a.u.) ± SE).

However, to the best of our knowledge, there is no evidence that such co-factors are involved in strain selection in the lymphoid tissue. At the molecular level, PrP$^{C}$ conformational landscape is likely to differ between spleen and brain, due to different trimming [40], glycosylation [64] including sialylation [65], and thus may impact conversion by certain PrP$^{Sc}$ subspecies, as observed during cross-species prion transmission. However, a parsimonious interpretation of our transmission experiments is that prion (sub)strain selection in the spleen relative to the brain is driven by PrP$^{C}$ expression levels. Here, LA19K / LA21K prions preferential targeting for brain / spleen from high expresser mice correlated with their capacity to dominantly replicate in mice expressing high / low PrP$^{C}$ levels. We also observed that a second pair of co-existing prion strains (T2$^{Ov}$ / T1$^{Ov}$) showing differing tropism for the spleen versus brain in high expresser mice showed a similar PrP$^{C}$-dependent selection in transgenic mice expressing variable PrP$^{C}$ levels, that is the most lymphotropic substrain preferentially replicated on low expresser mice and the most neurotropic on high expresser mice.

It may be noted that PrP$^C$ expression levels are classically quantified in the whole spleen and in the whole brain, yet meaning that different variations in PrP$^C$ levels could occur locally in prion-replicating cells [66]. The exact contribution of FDC to spleen PrP$^C$ expression level is unknown since these cells only account for 0.1–1% of the cells, making isolation and quantification of their PrP$^C$ expression levels a highly challenging issue. However, we previously reported by immunofluorescence studies that PrP$^C$ expression levels appeared low in tg338 FDC compared to neurons [21].

As a tentative explanation for strain-dependent polymerization rate as a function of PrP$^C$ levels, we inferred by mathematical modelling that differences in the number of PrP$^C$ monomers integrated in the growing PrP$^{Sc}$ assemblies (different kinetic order of the templating process) could explain PrP$^C$-dependent substrain selection [29]. LA19K / T2$^{Ov}$ growth would necessitate integration of more PrP$^C$ monomers than for LA21K / T1$^{Ov}$. In theory, the 'disadvantaged' strain that necessitates more PrP$^C$ to replicate should also start replicating in the spleen of high expresser mice or in the brain of low expresser mice. We show here a more contrasting situation. LA19K prions were barely able to replicate in high expresser spleens; T2$^{Ov}$ prions contained in tg338-adapted MM2-sCJD were eliminated by a single passage on low expresser mice. Exhaustion of PrP$^C$ substrate due to limiting cellular resource [67] or host-induced reduction of PrP$^C$ synthesis [68] by replication of the substrain needing less PrP$^C$ may prevent the chance that the kinetically-disadvantaged substrain replicates. Interference between substrains, which we suspected between T1$^{Ov}$ and T2$^{Ov}$ prions in certain brain regions of high expressers [28] may also prevent expression of T2$^{Ov}$ in low expresser mice. Irrespective of the molecular mechanisms, the outcome,—that is the selection of a unique minor component -, is reminiscent of that observed during cross-species transmission events (for reviews [7, 8]). PrP$^C$ level variations may thus drastically drive within-host prion diversification/evolution during homotypic transmission events.

Analyzing the disease tempo and which substrain among LA19K / LA21K was phenotypically dominant after IP infection provided some clues about the factors controlling prion neuroinvasion from the periphery. Typically, the IP route of inoculation favors neuroinvasion of prions replicating in the lymphoid tissue (review [43]). Accordingly, IP inoculation of LAN, tg338-passaged LAN (up to the 4[th] passage) or CH1641-like isolates allowed dominant accumulation of LA21K PrP$^{res}$ in the brain. Yet, a substantial proportion of the brains tested 19K-PrP$^{res}$ positive or PrP$^{res}$-negative at end life. However, LA21K prion replication is not kinetically disadvantaged in high expresser brains *per se* (Fig 3). In addition, the incubation time lengthening when IP and IC routes of infection were compared was overtly prolonged as compared to PG127 in tg338 mice or other experimental models. Quantitatively, a similar prolongation was observed after IP inoculation of cloned LA19K prions. Oppositely, IP inoculation of LAN404 sheep spleen, which appeared enriched in LA21K prions compared to sheep brain, showed faster pathogenesis with all brain analyzed 21K-PrP$^{res}$ positive (Fig 5). The events limiting the capacity and rapidity of LA21K prions to establish at full attack rate in the brain seem thus associated with the co-existence of LA19K/LA21K in the inoculum, LA19K prions imposing the disease tempo.

It remains to be determined why the IP ID was prolonged for cloned LA19K prions as a direct neuroinvasion process does not retard *per se* the disease tempo [19, 20]. PrP$^C$-limiting events may occur during LA19K journey to the brain, either in peripheral nerves or in the spinal cord.

In human infected with CJD, 21K and 19K PrP$^{Sc}$ signatures are detected in the same brain and are potentially associated with different strains [52, 53, 69]. The phenotypic spectrum of CJD, including for iatrogenic forms acquired peripherally, is highly heterogeneous (review [70]). Our data illustrate how the relative proportion of co-existing substrains could

profoundly influence the disease tempo and the pathological characteristics of the disease in infected hosts.

## Methods

### Ethics statement

Animal care and experiments were conducted in strict compliance with ECC and EU directives 86/009 and 2010/63. They were reviewed and approved by the local ethics committee of the author's institution, (name: COMETHEA: Comité d'Ethique en Expérimentation Animale du Centre INRA de Jouy-en-Josas et AgroParisTech). The permit numbers delivered by the COMETHEA are 12/034 and 15/056.

### Transgenic mice

As high expresser mice, we used tg338 mice that overexpress the VRQ allele of ovine PrP. The transgene construct consists of a large DNA sequence derived from ovine BAC libraries, encompassing natural regulatory sequences of the PrP gene transcription unit [71]. The PrP$^C$ levels in the brain are ~8-fold higher than in the sheep brain [29]. The spleen-to-brain PrP$^C$ ratio (~1:20) in tg338 mice is comparable to that found in conventional mouse models [21]. The strongest PrP$^C$ staining in tg338 spleens is FDC-associated [21]. Thus, there is no aberrant expression of PrP$^C$ in the spleen of these mice, quantitatively or qualitatively. As low expresser mice, we used tg335$^{+/-}$ (same transgene construct as the tg338 mice) and tg143$^{+/-}$ mice, which express 1.2-fold and 1.5-fold PrP$^C$ as compared to sheep brain, respectively [29, 71].

### Sheep TSE sources

Brain and spleen from sheep terminally affected with natural scrapie (LAN404 isolate) were provided by the Institut National de la Recherche Agronomique (O. Andréoletti, Toulouse, France). Brain extracts from French, Dutch, Spanish, Irish, and English sheep scrapie isolates were provided by the Institut National de la Recherche Agronomique (O. Andréoletti, Toulouse, France), the Central Veterinary Institute (J.M. Langeveld, Wageningen, Lelystad, Netherlands), the French National TSE Reference Laboratory (T. Baron, Anses, Lyon, France), CISA-INIA (J.M. Torres, Madrid, Spain), the Central Veterinary Research Laboratory, (E. Monks, Dublin, Ireland) and the former European TSE Reference Laboratory (J. Spiropoulos, VLA, Addlestone, UK). The CH1641 sheep scrapie strain [36] was provided by the Institute for Animal Health (N. Hunter, Edinburgh, UK). The TSE isolates are detailed in S1 Table.

### Tg338-passaged prion sources

Tg338-passaged LAN prions were obtained by iterative passage of brain material by IC route. The LAN404 isolate served as primary isolate. In the study, the 3$^{rd}$ passage was used unless indicated otherwise. Cloned LA19K prions were obtained by bicloning by limiting dilution in tg338 mice [29].

The T1$^{Ov}$ and T2$^{Ov}$ strains were co-isolated after serial transmission of a human sporadic CJD brain (MM2, rare cortical form) to ovine PrP tg338 mice [28]. As inoculum, a pool of brains from tg338 mice at the 4$^{th}$ serial passage of MM2-CJD and containing T1$^{Ov}$ and T2$^{Ov}$ prions, with T2$^{Ov}$ in higher proportion, was used. For comparison, pools of brains from tg338 mice inoculated with either cloned T2$^{Ov}$ prions obtained by bicloning by limiting dilution in tg338 mice or cloned T1$^{Ov}$ prions obtained by PMCA selection and single cloning [28] were used.

The characterization of LA21K *fast* and 127S strains in tg338 has been described previously [6, 29, 35, 71]. Pools of infected mouse brains were used as inoculum.

## Mouse transmission assays

Sheep tissue extracts were prepared as 10% w/v homogenate in 5% w/v glucose with a Precellys rybolyzer (Ozyme, Montigny-le-Bretonneux, France). To avoid any cross-contamination, a strict protocol based on the use of disposable equipment and preparation of all inocula in a class II microbiological cabinet was followed. For intracerebral inoculations, twenty microliters were inoculated in the right hemisphere to groups of individually identified mice, at the level of the parietal cortex. For intraperitoneal inoculations, hundred microliters of a 2% w/v solution in 5% glucose were used. For subsequent passage, mouse brains and spleens were collected with dedicated, sterile, disposable tools, homogenized at 20% w/v in 5% glucose; twenty microliters were reinoculated intracerebrally at 10% w/v. Animals were supervised daily for TSE development. Animals at terminal stage of disease or at end life were euthanized. To study the kinetics of PrP$^{Sc}$ accumulation in the spleen and in the brain after intraperitoneal or intracerebral infection, mice were euthanized healthy in triplicates at regular time-points post-inoculation, as indicated. For immunoblot analyses, brains and spleens were immediately frozen at -80˚C until use. For histoblot analyses, the collected brains were frozen on dry ice before storage at -80˚C.

## Immunoblot analyses

Brains and spleens were analyzed for proteinase K (PK)-resistant PrP$^{Sc}$ (PrP$^{res}$) content using a previously published protocol [28]. Briefly, PrP$^{res}$ was extracted from 20% w/v tissue homogenates with the Bio-Rad TeSeE detection kit. Aliquots were digested with PK (200 μg/mL final concentration) for 10 min at 37˚C before B buffer precipitation and centrifugation at 28,000 × g for 15 min. Pellets were resuspended in Laemmli sample buffer, denatured, run on 12% Bis/Tris gels (Bio-Rad), electrotransferred onto nitrocellulose membranes, and probed with 0.1 μg/mL biotinylated anti-PrP monoclonal antibody Sha31 antibody (human PrP epitope 145–152, [72]) or with 0.1 μg/mL anti-PrP 12B2 antibody (human PrP epitope 89–93, epitope, [42]) and followed by streptavidin conjugated to horseradish peroxidase (HRP) or by HRP conjugated to goat anti-mouse IgG1 antibody (1/20 000 final dilution), respectively. Immunoreactivity was visualized by chemiluminescence (GE Healthcare). For quantification, the total amount of PrP$^{res}$ or the relative amounts of PrP$^{res}$ glycoforms were determined by the use of GeneTools software after acquisition of chemiluminescent signals with a GeneGnome digital imager (Syngene, Frederick, MD).

## Histoblot analyses

Brain cryosections were cut at 8–10 μm, transferred onto Superfrost slides and kept at -20˚C until use. Histoblot analyses were performed as described [73], using the 12F10 anti-PrP antibody (human PrP epitope 142–160, [74]). Analysis was performed with a digital camera (Coolsnap, Photometrics) mounted on a binocular glass (SZX12, Olympus). The sections presented are representative of the analysis of three brains samples.

## Resistance to proteinase K digestion

20% (w/v) brain homogenates from tg338 mice infected with 127S and LA19K prions were diluted at 5% in a solubilization buffer (20 mM HEPES pH 7.4, 150 mM NaCl, 5 mM EDTA, 1 mM DTT, 2% (w/v) dodecyl-β-D-maltoside (Sigma)) 2% N-lauryl sarcosine (Fluka), final

concentrations) and digested for 2h at 37˚C with increasing concentrations of PK (0 to 10 000 µg/mL), as indicated. After digestion, the samples were diluted in equal volumes of Laemmli buffer, denatured and analyzed for PrP content by western blot, as above.

## PrP^Sc degradation by primary cultured peritoneal macrophages

To induce the multiplication of peritoneal macrophages, healthy tg338 mice were intraperitoneally injected with 3% Brewer thioglycolate broth (BD Biosciences). Three days after the injection, mice were euthanized by cervical column disruption. Peritoneal lavage was performed with 5 mL D-PBS (Gibco). The lavage fluid was mixed with an equivalent volume of 4˚C Dulbecco's Modified Eagle Medium (DMEM, Lonza) supplemented with 10% fetal calf serum (FCS, Biowhitaker), streptomycin and penicillin (PS, Gibco) before centrifugation at 100 x g for 5 min. Red cells were lyzed with hematolytic medium (155 mM ammonium chloride, 12 mM sodium carbonate, pH 7.4). The cells were resuspended in DMEM-10% FCS-PS and washed twice by centrifugation. After a live cell count, the cells were aliquoted in 6-well plates ($6.10^6$ cells / well) and incubated at 37˚C. After 24h, non-adhering cells were discarded by washing the wells with D-PBS. Macrophages were exposed to brain homogenates from tg338 mice infected with 127S and LA19K prions (1% (w/v) dilution in DMEM-10% FCS-PS) for 24h at 37˚C. After two washes in D-PBS, the cells were lyzed or incubated for 9 days. At regular time points, the contents of the wells were lyzed in lysis buffer (0.5% sodium deoxycholate, 0.5 Triton X-100, 50 mM Tris-HCl, pH 7,4) and centrifuged for 1 min at 500 x g. The supernatants were collected and analyzed for protein content (MicroBCA kit, Pierce). The equivalent of 250 µg of proteins was digested by 1µg of PK for 1h at 37˚C. The samples were then methanol precipitated, resuspended in Laemmli buffer, denatured and analyzed for PrP^res content by western blot.

## Supporting information

**S1 Fig. PrP^res electrophoretic profile from CH1641-like isolates.** Electrophoretic pattern of PrP^res in the brain from CH1641-like cases, compared with CH1641 isolate [36], PG127 isolate [71] and sheep experimentally inoculated with BSE prions [76].
(TIF)

**S2 Fig. Low levels of infectivity in the spleen of tg338 mice infected with cloned LA19K prions.** Cloned LA19K prions were obtained by serial passage of one LAN isolate (LAN404) and bicloning by limiting dilution in tg338 mice [29]. Brain or spleen extracts from tg338 mice infected with cloned 19K prions were inoculated (IC route) to reporter tg338 mice. Transmission with brain or spleen extracts are indicated with black and red lines, respectively. The number of affected/inoculated mice (mice with TSE and positive for brain PrP^res by immunoblot) and the mean survival times in days ± SEM are indicated for each inoculated group. Segmented, doubled circles are used to indicate the proportion of mice with 19K PrP^res signature (blue), 21K PrP^res signature (grey) or absence of PrP^res (white), either in the brain (inside of the circle, black lines) or in the spleen (outside of the circle, red lines). The data shown are representative of 5 independent transmission experiments with different mice infected with cloned LA19K prions. Data in italic are from [29].
(TIF)

**S3 Fig. Strain phenotype of prions replicating in tg338 mouse spleens on serial passage of LAN or CH1641-like isolates.** Transmissions by IC route of brain or spleen extracts from tg338 mice infected with the LAN isolate (99–378 isolate, 2^nd passage) or the CH1641-like isolate (O100 isolate, 2^nd passage) to reporter tg338 mice. Transmission with brain or spleen

extracts are indicated with black and red lines, respectively. The number of affected/inoculated mice (mice with TSE and positive for brain PrP$^{res}$ by immunoblot) and the mean survival times in days ± SEM are indicated for each inoculated group. Segmented, doubled circles are used to indicate the proportion of mice with 19K PrP$^{res}$ signature (blue) or 21K PrP$^{res}$ signature (grey) in the brain (inside of the circle, black lines) and the spleen (outside of the circle, red lines).

(TIF)

**S1 Table. TSE sources transmitted to tg338 mice.**
(PDF)

**S2 Table. PrP$^{res}$ signature in the brain and spleen after intraperitoneal inoculation of LAN, CH1641-like isolates and LA19K prions to tg338 mice.**
(PDF)

## Acknowledgments

We thank the staff of Animalerie Rongeurs (IERP, INRA, 2018. Infectiology of fishes and rodent facility, doi: 10.15454/1.5572427140471238E12, Jouy-en-Josas, France) for animal care, N. Moriceau for excellent technical help, P. Clayette (Bertin Pharma, Montigny-le-Bretonneux, France) and J. Langeveld (Wageningen University, Lelystad, The Netherlands) for the kind gift of Sha31 and 12B2 anti-PrP monoclonal antibodies, respectively. We thank our colleagues for kindly providing sheep TSE sources.

## Author Contributions

**Conceptualization:** Vincent Béringue, Human Rezaei, Hubert Laude.

**Data curation:** Vincent Béringue, Philippe Tixador, Olivier Andréoletti, Fabienne Reine, Johan Castille, Thanh-Lan Laï, Annick Le Dur, Aude Laisné, Laetitia Herzog, Bruno Passet, Human Rezaei, Jean-Luc Vilotte, Hubert Laude.

**Formal analysis:** Vincent Béringue, Olivier Andréoletti, Johan Castille, Thanh-Lan Laï, Annick Le Dur, Aude Laisné, Laetitia Herzog, Bruno Passet, Jean-Luc Vilotte, Hubert Laude.

**Funding acquisition:** Vincent Béringue, Human Rezaei, Jean-Luc Vilotte, Hubert Laude.

**Investigation:** Philippe Tixador, Olivier Andréoletti, Fabienne Reine, Johan Castille, Thanh-Lan Laï, Annick Le Dur, Aude Laisné, Laetitia Herzog, Human Rezaei, Jean-Luc Vilotte, Hubert Laude.

**Methodology:** Vincent Béringue, Philippe Tixador, Olivier Andréoletti, Johan Castille, Thanh-Lan Laï, Annick Le Dur, Aude Laisné, Laetitia Herzog, Bruno Passet, Human Rezaei, Jean-Luc Vilotte, Hubert Laude.

**Project administration:** Vincent Béringue.

**Resources:** Olivier Andréoletti, Johan Castille, Bruno Passet, Jean-Luc Vilotte.

**Supervision:** Vincent Béringue, Annick Le Dur.

**Validation:** Vincent Béringue, Philippe Tixador, Olivier Andréoletti, Fabienne Reine, Johan Castille, Thanh-Lan Laï, Annick Le Dur, Aude Laisné, Laetitia Herzog, Bruno Passet, Human Rezaei, Jean-Luc Vilotte, Hubert Laude.

**Writing – original draft:** Vincent Béringue.

**Writing – review & editing:** Vincent Béringue, Jean-Luc Vilotte, Hubert Laude.

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
