## [Decision Letter · Decision Letter 0]

31 Jan 2020

Dear Dr. Beringue,

Thank you very much for submitting your manuscript "Host prion protein expression levels impact prion tropism for the spleen" for consideration at PLOS Pathogens. As with all papers reviewed by the journal, your manuscript was reviewed by members of the editorial board and by several independent reviewers. In light of the reviews (below this email), we would like to invite the resubmission of a significantly-revised version that takes into account the reviewers' comments.

Specifically, two of the reviewers commented that the paper was difficult to penetrate, one to the extent it was difficult to understand and appraise even what had been done thus far to reach a preliminary verdict. In one instance there was also a question as to the degree of advance over prior contributions from this lab

Your revised manuscript will need to be sent to reviewers for further evaluation to see if a majorly restructured manuscript satisfies the issues raised by the first version. Unfortunately we cannot make any decision about publication until we have seen the revised manuscript, your response to the reviewers' comments and the reviewer's opinions of said responses.

Sincerely,

David Westaway

Section Editor

PLOS Pathogens

Neil Mabbott

Section Editor

PLOS Pathogens

Kasturi Haldar

Editor-in-Chief

PLOS Pathogens

orcid.org/0000-0001-5065-158X

Michael Malim

Editor-in-Chief

PLOS Pathogens

orcid.org/0000-0002-7699-2064

Reviewer's Responses to Questions

**Part I - Summary**

Reviewer #1: The authors previously shown that PrPC expression level in transgenic mice can influence the emergence of prion strains. The authors build upon these findings and provide evidence suggesting that differences in PrPC expression between the CNS and LRS also can influence strain tropism and emergence. The authors include an impressive body of transmission data and accompanying biochemical analysis to support this finding. Similar findings were observed using the scrapie and CDJ isolates strengthens the findings in the transgenic mouse system. Overall the conclusions are supported by the data. The discussion is thoughtful and the suggestion that the inclusion of PrPSc properties from spleen when stain typing is significant. Suggestions to improve the manuscript would be to include quantification and statistical analysis of PrPSc levels where conclusions are based on differences in abundance in figure 1D and Figure 6. The authors should double check the formatting of the references, some errors were noted.

Reviewer #2: The goal of this study by Beringue et al. was to uncover the underlying mechanism of strain tropism and selective replication of substrains in spleen and brain tissue, respectively. They propose that the PrPC expression level, which is about 20fold lower in the spleen than in the brain, determines replication, and demonstrate that in the spleen of transgenic mice overexpressing ovine PrP (tg338) LA21K prions are found, whereas in the brains of those mice mostly LA19K prions replicate. The same group has published earlier that strain selection in the brain is linked to PrPC expression levels, with LA21K replicating in the brains of low/physiological-level PrPC-expressing mice, while propagation of LA19K required high expression levels of PrPC (LeDur et al., Nature Communications 2017). The results of the current study re-iterate the importance of PrPC expression levels for the replication of specific substrains. However, it does not address other potential factors that might drive tissue tropism (eg. tissue-specific co-factors or PrPC modifications, conformational differences of PrPSc).

Reviewer #3: In this manuscript the authors attempt to discriminate and characterize two ovine strains through subsequent passages in transgenic (Tg) mice overexpressing sheep prion protein (PrP). The biochemical and neuropathological characterization and passaging strategies have identified a putative correlation with the expression levels of PrP in either brain or spleen.

**Part II – Major Issues: Key Experiments Required for Acceptance**

Reviewer #1: Suggestions to improve the manuscript would be to include quantification and statistical analysis of PrPSc levels where conclusions are based on differences in abundance in figure 1D and Figure 6.

Reviewer #2: The manuscript overall was difficult to follow and it was not well justified why so many different isolates were used, or even different passages of LA19K (Fig. 3C). PMCA would have been helpful to compare replication rates in the different tissues. Some statements are not supported by the results that are shown:

p. 13, lanes 4, 5 and Figure 1D: it is difficult to verify that accumulation of 21K-PrPres plateaued from 60 days post infection in the spleen. There is a clear increase to day 100, and at the terminal stage twice the amount of protein was loaded; the same is true for the panels showing brain PrPres, similar amounts of proteins should be loaded to allow comparison of propagation dynamics.

Figure 2B: the bands detected in brain and spleen from spleen-inoculated mice (lanes 3 and 4) is described as 21K-PrPres, however, it migrates faster than the 21K-PrPres in the brains of tg143+/-. Confirmation with 12B2 antibody would be helpful.

Figure 2C: the background signal in the lower panel is much higher, making it difficult to compare the appearance of PrPSc deposits.

Figure 4B: the PrPres band in lane 5 (i.c./Sp) is supposed to be 21K but appears to migrate faster than the 21kDa-pattern in lane 1; it is stated on p. 17/lane 14 that i.p. inoculation resulted in 21K-PrPres in brains and spleens, but the migration pattern in lanes 2 and 3 differs; re-probing with 12B2 might clarify the banding pattern. Further, the legend and figure annotation for lane 1 needs to be clarified.

Figure 6: The heading of the legend does not fit with the results shown. The statement that PK resistance and degradation rate of LA19K and 127S (what is the source of 127S?) are not different is not supported by the data. LA19K is detectable even with 10 mg/ml of PK and might be more resistant even to macrophage degradation, with a faint signal after 10 days. Quantification and statistical analysis is needed to draw a conclusion.

Reviewer #3: I had a very difficult time in reading this account. I think the data are solid and support the conclusion drawn in the discussion but it is a daunting task to actually understand what the authors sometime want to express in their writing.

I would like to suggest to completely re-write the manuscript and streamline the findings in an more methodical and systematic way.

This amount of experimental data deserve a clearer account of the findings and conclusions.

**Part III – Minor Issues: Editorial and Data Presentation Modifications**

Reviewer #1: The authors should double check the formatting of the references, some errors were noted.

Reviewer #2: Figure 6: The heading of the legend does not fit with the results shown.

p. 17, lane 14: results shown in Figure 4A - B are described, the authors refer to Figure 3A-B though.

Reviewer #3: Figures are difficult to follow. Too much information in a very short space. I would like to suggest to revise the presentation of results.

PLOS authors have the option to publish the peer review history of their article (what does this mean?). If published, this will include your full peer review and any attached files.

Reviewer #1: No

Reviewer #2: No

Reviewer #3: No
---

## [Decision Letter · Decision Letter 1]

22 Jun 2020

Dear Dr. Beringue,

We are pleased to inform you that your manuscript 'Host prion protein expression levels impact prion tropism for the spleen' has been provisionally accepted for publication in PLOS Pathogens.

Best regards,

Neil A. Mabbott

Section Editor

PLOS Pathogens

Neil Mabbott

Section Editor

PLOS Pathogens

Kasturi Haldar

Editor-in-Chief

PLOS Pathogens

orcid.org/0000-0001-5065-158X

Michael Malim

Editor-in-Chief

PLOS Pathogens

orcid.org/0000-0002-7699-2064

Reviewer Comments (if any, and for reference):

Reviewer's Responses to Questions

**Part I - Summary**

Reviewer #1: The authors have addressed the concerns of the reviewer.

Reviewer #2: This revised manuscript has been significantly improved by adding additional data and quantification of Western blot, and significant re-writing has enhanced clarity of the manuscript. The authors have convincingly addressed all major concerns.

**Part II – Major Issues: Key Experiments Required for Acceptance**

Reviewer #1: (No Response)

Reviewer #2: (No Response)

**Part III – Minor Issues: Editorial and Data Presentation Modifications**

Reviewer #1: (No Response)

Reviewer #2: (No Response)

PLOS authors have the option to publish the peer review history of their article (what does this mean?). If published, this will include your full peer review and any attached files.

Reviewer #1: No

Reviewer #2: No

---

## [Editor Report · Acceptance letter]

16 Jul 2020

Dear Dr. Beringue,

We are delighted to inform you that your manuscript, "Host prion protein expression levels impact prion tropism for the spleen," has been formally accepted for publication in PLOS Pathogens.

Best regards,

Kasturi Haldar

Editor-in-Chief

PLOS Pathogens

orcid.org/0000-0001-5065-158X

Michael Malim

Editor-in-Chief

PLOS Pathogens

orcid.org/0000-0002-7699-2064